# Strength of the Association of Elevated Vitamin B12 and Solid Cancers: An Adjusted Case-Control Study

**DOI:** 10.3390/jcm9020474

**Published:** 2020-02-09

**Authors:** Geoffrey Urbanski, Jean-François Hamel, Benoît Prouveur, Cédric Annweiler, Alaa Ghali, Julien Cassereau, Pierre Lozac’h, Christian Lavigne, Valentin Lacombe

**Affiliations:** 1Department of Internal Medicine, Angers University Hospital, 49933 Angers, France; benoit.prouveur@laposte.net (B.P.); alghali@chu-angers.fr (A.G.); pierre.lozac@gmail.com (P.L.); chlavigne@chu-angers.fr (C.L.); lacombe.valentin31@gmail.com (V.L.); 2Department of Biostatistics and Methodology, Angers University Hospital, 49933 Angers, France; jeanfrancois.hamel@chu-angers.fr; 3Department of Geriatric Medicine and Memory Clinic, Angers University Hospital, 49933 Angers, France; ceannweiler@chu-angers.fr; 4Department of Neurology, Angers University Hospital, 49933 Angers, France; jucassereau@chu-angers.fr

**Keywords:** vitamin B12, neoplasms, neoplasm metastasis, case-control study, biomarkers

## Abstract

The association between elevated plasma vitamin B12 (B12) level and solid cancers has been documented by two national registries. However, their design did not allow for the adjustment for other conditions associated with elevated B12. The objectives of this study were to confirm this association after the adjustment for all causes of elevated B12, and to study the variations according to the increasing B12 level, the type of cancers, and the presence of metastases. We compared 785 patients with B12 ≥ 1000 ng/L with 785 controls matched for sex and age with B12 < 1000 ng/L. Analyses were adjusted for the causes of elevated B12: myeloid blood malignancies, acute or chronic liver diseases, chronic kidney failure, autoimmune or inflammatory diseases, and excessive B12 supplementation. A B12 ≥ 1000 ng/L was associated with the presence of solid cancer without metastases (OR 1.96 [95%CI: 1.18 to 3.25]) and with metastases (OR 4.21 [95%CI: 2.67 to 6.64]) after adjustment for all elevated B12-related causes. The strength of the association rose with the increasing B12 level, in particular in cases of metastases. No association between liver cancers and elevated B12 level was found after adjustment for chronic liver diseases. In conclusion, unexplained elevated B12 levels should be examined as a possible marker of solid cancer.

## 1. Introduction

Vitamin B12 (B12) or cobalamin is essential for cell division [1]. Although its assay is mainly intended to detect deficiencies, incidental findings of elevated B12 levels are not uncommon [2]. In the absence of a consensus on the threshold that defines an abnormal increased B12 level, elevated B12 is generally defined as a level higher than the upper limit of the reference range, i.e., around 1000 +/− 100 ng/L (738 +/− 73.8 pmol/L) [2,3,4]. Abnormally elevated B12 has been attributed many causes [5,6], e.g., myeloid blood malignancies [7,8,9], acute or chronic liver diseases [10,11,12], chronic kidney failure, autoimmune or inflammatory diseases [3], and Gaucher disease [13].

A link between elevated vitamin B12 level and solid cancers has been discussed since the 1970s in case series, raising the possibility of using B12 as a biomarker of solid cancers [6]. Recently, two retrospective cohorts based on health registries have shown an association between elevated B12 and a newly-diagnosed solid cancer [14,15]. A B12 > 800 pmol/L (1084 ng/L) in the Danish registry published in 2013 and a B12 between 800 and 1000 pmol/L (1084–1355 ng/L) in the British registry published in 2019 were associated with the occurrence of a cancer within one year after the assay (Standardized Incidence Ration (SIR) respectively 6.3 [95%CI: 5.7–6.9] and 2.9 [95%CI: 2.4–3.5]) [15]. Nevertheless, in these registries, the association between elevated B12 and solid cancers was not adjusted for all elevated B12-related causes, except for excessive vitamin B12 supplementation in the Danish registry and alcohol misuse in the British registry.

In the Danish registry, a B12 > 800 pmol/L (1084 ng/L) was associated with the following cancer sites: pancreas, esophagus and stomach, colon and rectum, lungs, kidneys, urinary bladder, and strongly with liver [14]. Nevertheless, the risk for liver cancers in registry studies were not adjusted for chronic liver diseases, which might increase B12 levels. Moreover, the metastatic status was not assessed in this registry. Other studies focusing on specific cancers have shown a strong association between elevated B12 levels and liver cancers (OR 3.3 [95%CI: 1.1–10.4] for hepatocellular carcinomas and OR 4.7 [95%CI: 1.2–17.9] for other liver cancers) [16], a moderate association with prostate cancer (OR 1.1 [95%CI: 1.0–1.2]) [17], and no association with breast cancer (OR 1.1 [95%CI: 0.9–1.4]) [18]. The data about lung cancer were contradictory [19,20].

Few studies evaluated the strength of the association between solid cancers and B12 according to the level of B12 elevation. To our knowledge, the highest threshold evaluated was 1000 pmol/L (1355 ng/L) in the British registry. In this study, the association between solid cancers and elevated B12 appeared to be stronger in people with a B12 > 1000 pmol/L than those with levels between 800 and 999 pmol/L [15]. Higher B12 levels can be observed and raise the question about a growing association with solid cancers.

The association between elevated B12 and solid cancers is of great value, because it could be used as biological marker for screening for solid cancers. This requires that this association be independent of other causes of elevated B12. The main objective of our study was to confirm the association between elevated B12 and solid cancers after adjustment for all other known causes of elevated B12. The secondary objectives were to evaluate this association according to cancer and metastatic sites, to the level of B12 elevation, and to determine the B12 threshold that maximizes the association between elevated B12 and solid cancers according to the presence or absence of metastases.

## 2. Materials and Methods

### 2.1. Ethics and Statement for Study Checklist

This study was approved by the Bioethical Committee of Angers University Hospital (n° 2018-12). We applied the strengthening the reporting of observational studies in epidemiology (STROBE) statement to observational studies.

### 2.2. Study Population

This case-control study included patients aged 18 and over, hospitalized between January 2007 and December 2016 at Angers University Hospital, in the departments of Internal Medicine, Geriatrics or Neurology, and who had at least one plasma B12 testing. Patients admitted in the Intensive Care Unit were excluded to avoid transient elevations of vitamin B12 due to hemodynamic, infectious, or metabolic changes.

The group ‘High-B12’ included all the patients with a B12 ≥ 1000 ng/L. The threshold of 1000 ng/L corresponds to the normal upper limit of our assay, as in previous studies [4,21,22,23]. The group ‘Normal-B12’ included patients who had B12 < 1000 ng/L and matched with patients of the High-B12 group according to sex, age, and the hospitalization department, at a 1:1 ratio. These patients were randomly selected from the subjects that met the matching criteria.

### 2.3. Plasma Vitamin B12 Assay

Plasma B12 assays were centralized in the biochemistry laboratory of Angers University Hospital. The plasma vitamin B12 was collected in EDTA tubes and determined by immunoassay by competition, with direct chemiluminescence. The tests were carried out on an immunoanalytical system ADVIA Centaur^®^ (Siemens Healthcare Diagnostics Inc. Tarrytown, NY, USA) with ADVIA Centaur VB12^®^ reagents. The reference values considered normal by the supplier ranged from 198 to 986 ng/L with a coefficient of variation of 1.3–4.1%. The upper limit for the assay without dilution was 2000 ng/L.

In the case of multiple testing for the same patient, the highest B12 level was conserved.

### 2.4. Collected Data

The following general data were collected: sex, age, serum vitamin B12 levels, and date of B12 assay. The elevated B12-related causes were collected: acute (defined by an acute elevation of transaminases to more than 2-fold the normal) or chronic (defined by a dysmorphic radiological appearance, biological signs of liver failure, and/or histology suggestive of cirrhosis) liver diseases [6,10,11,12], severe chronic kidney failure (defined by an MDRD clearance of ≤30 mL/min/1.73 m²) [24], autoimmune or inflammatory diseases [3], myeloid malignancies [7,8,9], Gaucher disease [13], and excessive vitamin B12 supplementation in the absence of pernicious anemia (oral supplements of >1000 µg/week for more than 3 months or injectable supplements of >1000 µg/month for more than 3 months). Thresholds for defining excessive B12 supplementation were based on the fact that an oral cyanocobalamine supplementation with 1000 µg/week (or injectable supplements of 1000 µg/month) is sufficient for most cases of B12 deficiency, except for pernicious anemia and some rare causes with limited absorption or trafficking of cobalamin [25]. Lymphoid blood malignancies were also collected as negative controls.

We only considered solid cancers diagnosed 24 months within or prior the B12 assay. Data concerning the primary site of cancer, metastatic sites, and the time of appearance of metastases were collected.

### 2.5. Statistical Methods

Qualitative data are presented in absolute values and percentages. Quantitative data are presented in medians and interquartiles. Alpha risk was set at 5%, and the odds ratio (OR) was presented with a confidence interval of 95% (95%CI). Multivariate analyses were adjusted for the elevated B12-related causes and lymphoid blood malignancies. The adjusted OR were noted as aOR. The covariate Gaucher disease was excluded due to a small sample size (*n* = 2). Analyses were carried out using STATA 13.1 (StataCorp, Stata Statistical Software. College Station, TX, USA) and SPSS 20.0 (IBM Corp, IBM SPSS Statistics for Windows. Armonk, NY, USA).

#### 2.5.1. Association Between Solid Cancers and High-B12 Group

Univariate comparisons were made using McNemar’s test and paired *t*-test. Multivariate analyses were carried out by conditional logistic regression. The variable used to test the difference between both High-B12 and Normal-B12 groups was the presence of a cancer with or without metastases.

#### 2.5.2. To Determine the Threshold of B12 Level Maximizing the Association Between an Elevated B12 and Solid Cancers According to the Presence or Absence of Metastases

The threshold of B12 that maximizes the association between elevated B12 and solid cancers according to the presence or absence of metastases was determined by an exploratory manner. We evaluated the variation of this association (odds ratio) by increasing progressively the B12 threshold in a range of values between 300 and 2000 ng/L. First, we started to examine the association between elevated B12 and solid cancers at the threshold of 300 ng/L. Then, we examined the association at the threshold of 301 ng/L, and so on, till reaching the B12 threshold 2000 ng/L. For each tested threshold, the OR represented the association between elevated B12 and solid cancers by comparing patients with B12 concentration above this threshold with those having B12 concentration below this threshold. These different analyses were adjusted for age, sex, and comorbidities (as detailed above). They were conducted separately according to the presence or absence of metastases, to determine the B12 threshold that maximizes the association between elevated B12 and solid cancers in each of these situations, and to evaluate the robustness of this association in these two situations. The graphical representation of the strength of the association between solid cancer and elevated B12 presents the progression of aOR according to all tested thresholds.

#### 2.5.3. The Association Between Elevated B12 and Solid Cancers by Interval of B12 Level According to the Metastatic Status

The strength of the association between solid cancers and elevated B12 was studied depending on the metastatic status and the level of B12 with the following intervals: <750, 750–999, 1000–1249, 1250–1749, and ≥1750 ng/L. The distributions were compared using the chi-squared test.

#### 2.5.4. The Association Between the High-B12 Group and Solid Cancers According to the Type of Cancer and the Primary Tumor and Metastatic Site

The association between the High-B12 group and the presence of a solid cancer was studied according to the primary cancer and metastatic site by conditional logistic regression. We have defined two independent statistical models adjusted for comorbidities: one with all the primary tumor sites in a first model, and another one with all the metastatic sites in a second model. Variables with less than five subjects per group were grouped under the category “other”.

## 3. Results

### 3.1. Population Description (Table 1)

The study population included 10,573 patients who underwent B12 assay between January 2007 and December 2016. We compared the 785 patients with B12 ≥ 1000 ng/L (7.4%) (High-B12 group) with 785 control subjects having normal level of B12 < 1000 ng/L (Normal-B12 group), matched according to age, sex, and department of hospitalization.

### 3.2. Causes of Elevated B12 in Multivariate Analysis

After adjustment, all causes of elevated B12 were more frequent in the High-B12 group, except for autoimmune or inflammatory diseases (Table 2). Solid cancers without or with metastases were associated with the High-B12 group, with aOR of 1.96 [95%CI: 1.18–3.25] and 4.21 [95%CI: 2.67–6.64], respectively.

### 3.3. Determination of the Best Thresholds of B12 Level to Evaluate the Association with Solid Cancers Based on the Presence or Absence of Metastases

After adjustment for age, sex, and comorbidities, the most discriminatory thresholds were 750 ng/L and 1000 ng/L for nonmetastatic and metastatic cancers, respectively (Figure 1, smoothed estimates). At a threshold of 750 ng/L, the aOR for the association between elevated B12 and solid cancers were 2.64 (95%CI: 1.59–4.37] and 3.48 (95%CI: 2.28–5.32]) for nonmetastatic and metastatic cancers, respectively. At a threshold of 1000 ng/L, the aOR for the association between elevated B12 and solid cancers were 2.03 [95%CI: 1.26–3.25] and 3.79 [95%CI: 2.50–5.74] for nonmetastatic and metastatic cancers, respectively.

### 3.4. Strength of the Association Between Solid Cancers and Elevated B12

#### 3.4.1. Based on the Metastatic Status of the Cancer by Plasma B12 Level Intervals (Table 3)

A significant association between elevated B12 and solid cancers was found from the interval of 750–999 ng/L in cases of nonmetastatic cancer and from the interval 1000–1249 ng/L in cases of metastatic cancer. The frequency of solid cancers increased with the B12 level (*p* < 0.0001), especially in case of metastatic cancers, and beyond 1250 ng/L: 7.5% for the <750 ng/L subgroup, 15.2% for the 750–999 ng/L subgroup, 13.7% for the 1000–1249 ng/L subgroup, 21.3% for the 1250–1749 ng/L subgroup, and 24.1% for the ≥1750 ng/L subgroup.

#### 3.4.2. According to the Primary Cancer and Metastatic Site (Table 4)

The following solid cancer sites were significantly associated with the High-B12 group: pancreas, urothelium, colon and rectum, lungs, and prostate. Liver cancers did not appear to be significantly associated with the High B12 group after adjustment for other causes of elevated B12 (aOR 1.5 [95%CI: 0.3–7.1]), especially chronic and acute liver diseases.

Liver and bone metastases were more frequent in the High-B12 group with aORs of 4.9 [95%CI: 2.2–11.0] and 3.1 [95%CI: 1.3–7.3], respectively.

## 4. Discussion

Studies that demonstrate an association between elevated B12 and solid cancers were based on either large registries with limited analyses of confounding factors, or on small retrospective cohorts focusing on a single type of cancer. The objective of our study, based on the analysis of 1570 medical files, was to consolidate the association between elevated vitamin B12 levels and solid cancers by adjusting for other known causes of B12 elevation. Additionally, the study aimed to evaluate the strength of this association according to the threshold of B12, the site of primary cancer, and the presence of metastases.

In line with the literature, our study demonstrated the association between elevated B12 and the different causes thereof, except for autoimmune and inflammatory diseases. Concerning blood malignancies, we found an association between elevated B12 and myeloid malignancies, but not with lymphoid malignancies, as previously described [7,26,27].

In our study, plasma vitamin B12 ≥ 1000 ng/L was observed in 7.4% of patients. This confirmed that the incidental finding of elevated B12 is not uncommon, even in departments with no specific recruitment of patients with liver or malignant diseases. This finding was slightly lower than that reported in previous studies, including among inpatients with frequencies ranging from 10 to 15% [3,16]. Lower prevalence of elevated B12 was observed in the general population: 3.5% of the population had a B12 > 813 ng/L in the British registry [15], and 6.6% of the population had a B12 > 1084 ng/L in the Danish registry [14]. The differences between these studies may be explained by variations in the recruitment (mean age, inpatients or outpatients, studied departments) and the threshold defining the status of elevated B12.

Our study confirmed the results of previous studies by showing an association between solid cancers and elevated B12. Moreover, we demonstrated that this association persisted after adjustment for other causes of elevated B12. In our study, the aOR was 2.0 [95%CI: 1.2–3.2] for nonmetastatic cancers and 4.2 [95%CI: 2.7–6.6] for metastatic cancers with a threshold of 1000 ng/L defining the elevated B12. Danish and British registries reported higher SIRs, i.e., 6.3 [95%CI: 5.7–6.9] and 4.7 [95%CI: 4.0–5.6], respectively [14,15]. There may be two explanations for these differences: the thresholds defining elevated B12 differed between the two registries (1084 and 1355 ng/L, respectively), and were higher than that of our study. However, we demonstrated that the frequency of solid cancers increased with B12 level. Another explanation could be the lack of adjustment for other causes that are known to be associated with elevated B12, which could artificially raise the strength of associations in the British and Danish registries.

In our study, cancer and metastatic sites associated with elevated B12 were almost the same as those found in registries’ studies: pancreas, esophagus and stomach, colon and rectum, lungs, kidneys, and urothelium. Liver cancer was also linked to elevated B12 levels, with a SIR of 11.4 [95%CI: 6.6–19.5] for a B12 levels between 800 and 1000 pmol/L in the British registry [15] and a SIR of 40.7 [95%CI: 25.5–61.6] for a B12 level >800 pmol/L in the Danish registry [14], whereas this association was not significant in our study. This difference could be explained by the absence of adjustment for liver diseases in the registries’ studies. In our study, no significant association was found between liver cancer and elevated B12 after considering the presence of acute and chronic liver diseases as covariates. This finding is in line with results of Simonsen et al. in 2014, who concluded that the elevation of vitamin B12 levels observed in liver cancers was linked to underlying chronic liver diseases [10].

The distribution of solid cancer sites in the Normal B12 group was comparable to the general French population [28]. The inclusion of patients hospitalized in the neurology department did not influence the number of patients with cerebral metastases. Our study included hospitalized patients who were older than those of the Danish and British registries, but this possible bias was alleviated by matching both groups of our study for age, sex, and hospitalization department.

Data concerning active smoking and alcohol misuse were collected: the proportion of alcohol misuse (78/785 in High-B12 group versus 59/785 in Normal-B12 group) and tobacco use (88/785 in High-B12 group versus 78/785 in Normal-B12 group) did not differ. However we decided not to adjust the analyses with these data. First, these data represent risk factors and are not comparable to conditions associated with elevated B12, which are established diseases. Secondly, the retrospective data collection constituted a risk of bias due to the possible difference of data reliability between patients with and without cancer.

To the best of our knowledge, looking for variations in the strength of association between B12 levels and solid cancers has been evaluated up to a threshold of 1355 ng/L in the British cohort [15]. In our study, we evaluated this association at higher thresholds. We showed that the association with cancers, especially metastatic ones, tended to increase with increasing B12 levels. This strong association between metastatic cancers and elevated B12 suggests a link between the importance of B12 level and the tumor mass or extension. The prognostic value of elevated B12 in solid cancers was already emphasized, and a one-year survival rate was reported in patients with a solid cancer of 62.3% in cases with B12 levels between 200 and 600 pmol/L, 49.6% between 600 and 800 pmol/L and 35.8% above 800 pmol/L (*p* < 0.0001) [29]. While the pathophysiological explanation of an elevated B12 for solid cancers is not elucidated, some authors think that B12 elevation could be secondary to the inflammation induced by the antitumor immune response, with plasma haptocorrin release (transcobalamins I and III) by the inflammatory cells [5,6,29]. However, a direct link with tumor mass is not excluded.

In the absence of a consensus on the threshold that defines an abnormal elevated B12, we chose the symbolic value of 1000 ng/L, defined as the upper normal limit of our test. This corresponds to the threshold that maximizes the association between solid cancer and elevated B12 in case of metastatic cancer. Nevertheless, we observed a nonmetastatic solid cancer risk at an interval of 750–1000 ng/L. These results raise the question about the relevance of the normal B12 levels. These results confirmed findings of the British registry in which an increased risk of solid cancers was noted from the interval 813–1084 ng/L (600–800 pmol/L).

In our study, an elevated B12 on a single testing was sufficient to classify the patient in the High-B12 group. Since the association between solid cancers and B12 levels was stronger between 1250 and 1749 ng/L or ≥1750 ng/L than in patients with B12 < 1250 ng/L, we cannot exclude the possibility that some acute situations could temporally increase B12 levels, biasing the association with solid cancers. This is why we excluded patients in intensive care units, and adjusted the analyses for acute liver diseases. Nevertheless, we cannot totally exclude the possibility of unknown disturbing factors in these hospitalized patients. It would be therefore interesting to assess the association between solid cancers and elevated B12 based on persistent elevated B12 levels. In the absence of therapeutic cancer management, increases in B12 would persist in cases of underlying cancer. The retrospective character of our study limited the interpretation of the time sequence of events, especially the causality link between solid cancers and elevated B12. To limit this possible bias, we limited the study for cancers diagnosed within the 24 months that followed or preceded the B12 testing with an active cancer at the time of the B12 assay, in order to only study cancers with a possible causality link with B12 elevation.

Although some authors have proposed guidelines to explore incidental B12 elevation, there is still no codified management [5,30,31]. From our point of view, in cases of incidental B12 elevation, clinicians should perform an extensive clinical examination, complete with simple additional tests (blood count, blood creatinine level, liver functions assessment, and liver echography) to eliminate the elevated B12-related causes other than solid cancers. In cases of unexplained B12 elevation, especially more than 1250 ng/L, we suggest the possible presence of an underlying solid cancer, with a special focus on cancer sites highly associated with elevated B12. Nevertheless, the relevance of extensive and expensive explorations remains uncertain, and strongly depends on the context. Another interesting question would be the evolution of vitamin B12 during the course and treatment of solid cancers.

## 5. Conclusions

An abnormal elevation of B12 was significantly associated with the presence of a solid cancer, even after an exhaustive adjustment for all the known causes of elevated B12. This association was significant for nonmetastatic cancers but stronger for metastatic cancers. Contrary to results from registries’ studies, we did not confirm the association of elevated B12 with liver cancers after adjustment for liver diseases. The solid cancer and metastatic sites most associated with elevated B12 were pancreas, colon/rectum, lungs, prostate, urothelium, and bone, and liver metastases.

## Figures and Tables

**Figure 1 jcm-09-00474-f001:**
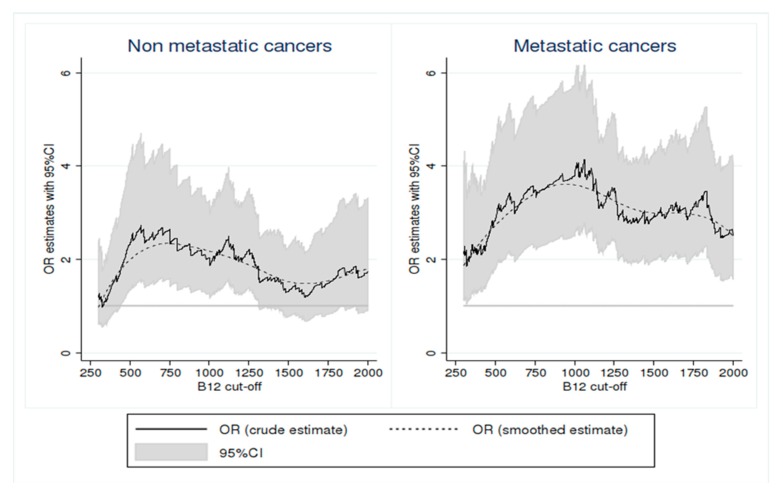
Strength of the association between solid cancers with and without metastases and elevated vitamin B12 levels. Notes: Graphics present the adjusted OR illustrating the association between solid cancer and elevated vitamin B12 levels for all values of vitamin B12 between 300 and 2000 ng/L. For each tested threshold, the OR represented the association between elevated B12 and solid cancers by comparing patients with B12 concentration above this threshold with those having B12 concentration below this threshold (and not with a single reference group of patients with normal B12). These two analyses are independent. For each analysis, cancer without metastases in the left panel and with metastasis in the right panel was defined as a dichotomous variable (presence versus absence). The odds ratios are adjusted for sex, age, chronic and acute liver diseases, severe chronic kidney failure, autoimmune and inflammatory diseases, excessive vitamin B12 supplementation, myeloid, and lymphoid blood malignancies.

**Table 1 jcm-09-00474-t001:** Comparison of Normal-B12 and High-B12 groups.

	Normal-B12 * (*n* = 785)	High-B12 ^†^ (*n* = 785)	*p*-value
**General characteristics**			
Age (years)	82.0 (73.0–89.0)	82.0 (73.0–89.0)	-
Sex (women)	472 (60.1%)	472 (60.1%)	-
Vitamin B12 (ng/L)	388 (277–534)	1408 (1147–1868)	<0.001
**Elevated B12-related causes**			
Chronic liver diseases	25 (3.2%)	54 (6.9%)	0.001
Acute liver diseases	20 (2.5%)	64 (8.2%)	<0.001
Severe chronic kidney failure	74 (9.4%)	128 (16.3%)	<0.001
Autoimmune/inflammatory diseases	68 (8.7%)	49 (6.2%)	0.09
Excessive vitamin B12 supplementation	3 (0.4%)	16 (2.0%)	0.004
Myeloid blood malignancies	8 (1.0%)	34 (4.3%)	<0.001
**Lymphoid blood malignancies**	25 (3.2%)	24 (3.1%)	>0.99
**Solid cancers**			
Without metastases	64 (8.2%)	152 (19.4%)	<0.001
With metastases	33 (4.2%)	100 (12.7%)	<0.001
**Site of primary cancer**			
Colon/rectum	10 (1.3%)	25 (3.2%)	0.02
Liver	3 (0.4%)	7 (0.9%)	0.34
Pancreas	3 (0.4%)	11 (1.4%)	0.06
Skin	2 (0.3%)	6 (0.8%)	0.29
Lungs	9 (1.1%)	16 (2.0%)	0.23
Prostate	13 (1.7%)	28 (3.6%)	0.03
Kidneys	5 (0.6%)	3 (0.4%)	0.73
Breast	11 (1.4%)	22 (2.8%)	0.06
Urothelium	4 (0.5%)	15 (1.9%)	0.01
Uterus (body/neck) and ovaries	4 (0.5%)	8 (1.0%)	0.38
Esophagus and stomach	2 (0.3%)	8 (1.0%)	0.11
Others	3 (0.4%)	22 (2.8%)	<0.001
**Location of metastases**			
Brain	7 (0.9%)	9 (1.1%)	0.79
Liver	10 (1.3%)	56 (7.1%)	<0.001
Bones	9 (1.1%)	44 (5.6%)	<0.001
Lungs	9 (1.1%)	28 (3.6%)	0.002
Lymph nodes	7 (0.9%)	30 (3.8%)	<0.001
Others	12 (1.5%)	25 (3.2%)	0.03

* B12 < 1000 ng/L. ^†^ B12 ≥ 1000 ng/L.

**Table 2 jcm-09-00474-t002:** Association between a plasma vitamin B12 level ≥ 1000 ng/L and solid cancers with and without metastases in multivariate analysis.

	Adjusted OR * [95% CI]	*p*-value
Sex (Males)	0.87 [0.70–1.08]	0.21
Age ^†^	0.93 [0.75–1.17]	0.55
Chronic liver diseases	2.80 [1.65–4.76]	<10^−4^
Acute liver diseases	2.43 [1.40–4.21]	0.002
Severe chronic kidney failure	2.03 [1.43–2.88]	<10^−4^
Autoimmune or inflammatory diseases	0.83 [0.56–1.23]	0.35
Excessive vitamin B12 supplementation	6.87 [1.98–23.86]	0.002
Myeloid malignancies	5.46 [2.44–12.25]	<10^−4^
Lymphoid malignancies	1.22 [0.65–2.30]	0.53
Cancer		
No cancer	1.0 (reference) ^‡^	-
Cancer without metastases	1.96 [1.18–3.25]	0.003
Cancer with metastases	4.21 [2.67–6.64]	<10^−4^

***** Adjustment for sex, age, chronic and acute liver diseases, chronic kidney failure, autoimmune and inflammatory diseases, excessive vitamin B12 supplementation, myeloid and lymphoid malignancies, and solid cancers. **^†^** Age as a categorical variable (< or ≥ 80 years). ^‡^ Cancer as a categorical variable with 3 outcomes: No cancer, Cancer without metastases, and Cancer with metastases.

**Table 3 jcm-09-00474-t003:** Solid cancers frequency according to vitamin B12 interval and to the presence of metastases.

Vitamin B12 (ng/L)	0–749	750–999	1000–1249	1250–1749	≥1750
No cancer (n, %)	665/719 (92.5%)	56/66 (84.8%)	246/285 (86.3%)	211/268 (78.7%)	176/232 (75.9%)
Cancer without metastases (n, %)	23/719 (3.2%)	8/66 (12.1%)	14/285(4.9%)	23/268 (8.6%)	15/232 (6.5%)
Cancer with metastases(n, %)	31/719 (4.3%)	2/66 (3.0%)	25/285(8.8%)	34/268 (12.7%)	41/232 (17.7%)

The distributions differed between the subgroups with vitamin B12 elevation (*p* < 0.0001).

**Table 4 jcm-09-00474-t004:** Association between plasma vitamin B12 ≥ 1000 ng/L and solid cancers according to cancer and metastatic sites in multivariate analysis.

	Adjusted OR * [95%CI]	*p*-value
**A—Sites of solid cancers ^†^**		
Colon and rectum	3.02 [1.35–6.75]	0.007
Liver	1.46 [0.30–7.15]	0.64
Pancreas	4.00 [1.02–15.65]	0.04
Skin	2.62 [0.49–13.92]	0.26
Lungs	2.89 [1.14–7.35]	0.03
Prostate	2.17 [1.02–4.63]	0.04
Kidneys	0.31 [0.06–1.66]	0.17
Breasts	1.86 [0.77–4.47]	0.17
Urothelium	7.40 [1.77–30.87]	0.006
Uterus (body/neck) and ovaries	1.07 [0.30–3.82]	0.91
Esophagus and stomach	3.47 [0.69–17.49]	0.13
Others	14.94 [3.89–57.3]	<10^−4^
**B—Sites of metastases ^†^**		
Brain	0.57 [0.14–2.30]	0.43
Liver	4.88 [2.16–11.00]	<10^−4^
Bones	3.11 [1.33–7.27]	0.009
Lungs	2.10 [0.76–5.82]	0.15
Lymph nodes	2.58 [0.94–7.07]	0.07
Others	0.59 [0.22–1.58]	0.29

***** Adjustment for sex, age, chronic and acute liver diseases, severe chronic kidney failure, autoimmune and inflammatory diseases, excessive vitamin B12 supplementation, myeloid and lymphoid blood malignancies. **^†^** The conditional logistic regression models of items A and B are independent. However, all variables listed under each item are included in the models as well as age, sex, and comorbidities.

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
