# Peer review of "Strength of the Association of Elevated Vitamin B12 and Solid Cancers: An Adjusted Case-Control Study"

_jcm, 2020, doi:10.3390/jcm9020474_

Round 1

Reviewer 1 Report

The authors have investigated the association between elevated vitamin B12 and solid cancer. They adjusted for know causes of elevated vitamin B12 and found that a vitamin B12 level above 1000 ng/L was associated with solid cancers with and without metastases.

I have a few remarks and minor corrections;

page 2 line 90; please add the precision, upper limit and reference values of the applied assay. How do you cope with samples that maximize the upper limit; where these samples diluted? Please add this information to the method section. page 3 line 100; excessive vitamin B12 supplemention was defined as usage of supplements or injections for more than 3 months. Please explain why the authors have chose a period of more than 3 months and explain how this data was collected (e.g. questionnaires?). Plasma vitamin B12 levels rises quickly after intake of high levels and I assume that if patients have started vitamin B12 intake for less than 3 months this would have affected your analysis. Please add a paragraph about this in the discussion page 2 line 135; how have the authors chosen the intervals? Were these for instance based on quartiles? Please explain. page 6 figure 1. In figure 1 the authors show the OR according to vitamin B12 levels. They found that at certain cut-offs the OR increases. Interestingly, the OR seems to decrease at a certain point. Would this mean that there might be a window at which vitamin B12 is associated with solid cancers and when levels become higher this is less strong? Could the authors test and discuss this in their paper as this might be important information. This might also relate to the fact that samples might be diluted due to the upper limit of the assay? page 7, table 4. The authors have examined the association of B12 with solid cancers for each separate sites. I would like to see the mean vitamin B12 levels of each of these sites to assess whether B12 levels increase more in urothelium than for instance lungs.  Bases on the OR I would expect this. Could the authors add a table or figure with this information?

Minor corrections;

page 2 line 86; do the authors mean concentration by dosage? Dosage is mostly related to intake and concentrations reflects what is measured in plasma page 2 line 87; please add which preservant was used to collect plasma (e.g. EDTA or heparine) page 5 line 153-154. The authors refer to table 2 but I think this data is present in table 1? page 6, table 3. I am not sure what this table adds. All relevant data are present to my opinion in figure 1. Maybe this table could be put in supplementals?

Author Response

Response to reviewer 1: revised manuscript jcm-714905

February, the 4th, 2020

Dear Editor,

Dear Reviewers,

                We wish to thank the reviewer for the analysis of our work and for the helpful comments. We are thankful to let us submitting a revised manuscript for this study that highlights the association between solid cancers and elevated plasma vitamin B12, even after adjustment for other known causes of elevated B12.

                Please find below the answers to the comments in order to clarify and improve the quality of our manuscript.

Reviewer 1:

Comment: The authors have investigated the association between elevated vitamin B12 and solid cancer. They adjusted for known causes of elevated vitamin B12 and found that a vitamin B12 level above 1000 ng/L was associated with solid cancers with and without metastases. I have a few remarks and minor corrections.

Response: We would like to thank the reviewer for the summary of our study and for the comments that helped us to improve the clarity of our manuscript.

Question 1: Page 2 line 90. Please add the precision, upper limit and reference values of the applied assay. How do you cope with samples that maximize the upper limit; where these samples diluted? Please add this information to the method section.

Response: For this test, the reference values considered as normal by the supplier ranged from 198 to 986 ng/L, with a coefficient of variation between 1.3% and 4.1% (H. Zhang et al. 69th AACC Annual Scientific Meeting. A458. 2017). The upper limit for the assay without dilution was 2000 ng/L. The linearity was respected up to this limit in our test. Above 2000 ng/L, the results were expressed as ≥2000 ng/L. This explained why we used the value of 2000 ng/L as the upper limit in our study. No dilution was performed for patients with a vitamin B12 level ≥2000 ng/L in this observational study. We added some precisions in the revised manuscript (pages 2-3 lines 90-92) and we thank the reviewer for this comment allowing more accuracy for the manuscript.

Question 2: Page 3 line 100. Excessive vitamin B12 supplementation was defined as usage of supplements or injections for more than 3 months. Please explain why the authors have chosen a period of more than 3 months and explain how this data was collected (e.g. questionnaires?). Plasma vitamin B12 levels rises quickly after intake of high levels and I assume that if patients have started vitamin B12 intake for less than 3 months this would have affected your analysis. Please add a paragraph about this in the discussion.

Response: The duration of supplementation was assessed on the base of medical records and prescriptions. In the study population with a median age of 82 years, we would like to precise that we had a high percentage of centralized follow-up that limited the risk to lose the information.

Concerning the threshold for the dose of vitamin B12 supplementation, an oral supplementation with 1000 µg of cyanocobalamine per week is sufficient for most cases of vitamin B12 deficiency (Andrès E. et al. The American Journal of Medicine 2001; 111:126–9). Except for pernicious anemia, the indications for high-dose oral or injectable vitamin B12 supplementation are rare (with limited absorption or trafficking of cobalamin). For this reason, we only mentioned pernicious anemia in the initial version of the manuscript as a cause of B12 deficiency that necessitates high B12 dosage (>1000µg/week).

Concerning the choice of duration to define excessive vitamin B12 supplementation, we were based on our own experience due to the fact that there was no consensual data in the literature. Indeed, in our clinical and research practices in the field of vitamin B12 deficiency (submitted manuscript), we did not observe elevation of plasma B12 over 1000 ng/L before 3 months of supplementation when the recommended dosages and indications were respected (1000 µg/day in pernicious anemia and 1000 µg/week in other causes of B12 deficiency for oral supplementation).

Concerning the risk of unknown vitamin B12 supplementation, we would like to underline the fact that vitamin B12 supplementation is normally prescribed in case of deficiency, which is a more common situation in the older patients. Thus, although that we cannot definitely exclude a hidden excessive supplementation (sometimes observed in some context, as body building for example), we considered that this risk was limited in our study population.

We added some precisions in the revised version of the manuscript (page 3, lines 103-106).

Question 3: Page 2 line 135. How have the authors chosen the intervals? Were these for instance based on quartiles? Please explain.

Response: We thank the reviewer for this relevant comment, which allowed us to clarify the stratification process of this analysis. We chose the first interval (0-750 ng/L) because the threshold of vitamin B12 level 750 ng/L was the most discriminatory threshold for non-metastatic cancers (Figure 1). The interval of 750-1000 ng/L was chosen because the threshold of 1000 ng/L was the most discriminatory threshold for metastatic cancers (Figure 1). The other intervals were fixed with the goal of maintaining similar or proportional ranges (250 ng/L or 500 ng/L) and enough statistical power. Moreover, we considered of interest to set up thresholds with rounded values that can be used by clinicians in daily practice.

Question 4: Page 6 figure 1. In figure 1 the authors show the OR according to vitamin B12 levels. They found that at certain cut-offs the OR increases. Interestingly, the OR seems to decrease at a certain point. Would this mean that there might be a window at which vitamin B12 is associated with solid cancers and when levels become higher this is less strong? Could the authors test and discuss this in their paper as this might be important information. This might also relate to the fact that samples might be diluted due to the upper limit of the assay?

Response: In the analysis represented in Figure 1, each value of vitamin B12 was considered as a potential threshold and did not represent the ‘risk’ of solid cancer for patients with this value compared to normal range. For example, when testing the threshold of 1750 ng/L in Figure 1, the OR represented the risk of solid cancer comparing patients with vitamin B12 ≥1750 ng/L to patients with vitamin B12 <1750 ng/L, and not with patients that had normal B12 level. As a consequence, the decreasing OR observed with the most elevated threshold in Figure 1 is explained by the increasing proportion of false negatives (i.e. patients with solid cancer and a plasma vitamin B12 level <1750 ng/L in this example). This analysis had double objective: 1) to determine if there was a threshold-effect for the association between plasma vitamin B12 and solid cancers (aspect of bell-shaped curves, as in our study), and 2) to identify these thresholds in cancer without or with metastases. We hope that this explanation justifies the interest of the Table 3, in which we compared the risk of cancer according to the increasing of B12 level with the reference group (vitamin B12 <750 ng/L). As shown with this interval analysis, the association between cancer and elevated B12 rose with levels of B12. We added precisions in the revised manuscript for more clarity (in the Methods section page 3 line 130-133, and in the notes of the figure 1 page 6 lines 186-190).

Question 5: Page 7, table 4. The authors have examined the association of B12 with solid cancers for each separate sites. I would like to see the mean vitamin B12 levels of each of these sites to assess whether B12 levels increase more in urothelium than for instance lungs.  Bases on the OR I would expect this. Could the authors add a table or figure with this information? 

Response: Median plasma vitamin B12 of each site of primary cancer are presented in the table below. We perfectly understand the interest of the reviewer for these data. However, we think that this approach differs from that we used all-over the manuscript: these median vitamin B12 levels are continuous raw data and cannot be easily adjusted for the other causes of elevated plasma vitamin B12. In our study, we used the fact to be below or above a threshold, as a categorical variable. Moreover, the fact that these continuous raw data for plasma vitamin B12 do not take into account the other elevated-B12 related causes, like chronic liver diseases, can be troubling for readers for example for the association with liver cancers, which is an important message of the article. We hope that the reviewer understands that we prefer not to add this data in the manuscript to avoid misleading the readers. However, we could add this table as supplementary material if the reviewer or the editor still found that this data needs to be specified.

Site of primary cancer

Vitamin B12 (ng/L, median and quartiles)

Colon/rectum

1118 [435-1647]

Liver

1243 [1105-2000]

Pancreas

1391 [752-1856]

Skin

925 [500-1435]

Lungs

1183 [616-1522]

Prostate

1106 [447-1457]

Kidneys

515 [265-1345]

Breast

1112 [457-1666]

Urothelium

1203 [1013-1838]

Uterus (body/neck) and ovaries

485 [316-1272]

 Esophagus and stomach

1139 [396-1567]

Others

1256 [1013-1838]

Question 6: Page 2 line 86; do the authors mean concentration by dosage? Dosage is mostly related to intake and concentrations reflects what is measured in plasma

Response: We meant assay by “dosage”. We apologize for this lack of written expression that we corrected in the revised version of the manuscript (page 2 line 86 “plasma B12 assays”, page 3 line 93 “B12 concentration”, and page 9 line 307 “assay”).

Question 7: Page 2 line 87. Please add which preservant was used to collect plasma (e.g. EDTA or heparine).

Response: We used EDTA tubes. We added this precision in the revised version of the manuscript (page 2, line 87).

Question 8: Page 5 line 153-154. The authors refer to table 2 but I think this data is present in table 1?

Response: We apologize for the lack of clarity but the tables refer to different results. The table 1 presents the univariate comparison between the High-B12 and the Normal-B12 groups, without adjustment. We agree that these comparisons are questionable relative to our first objective but we wanted to present the characteristics of the 2 groups for the readers before presenting multivariate results. On the contrary, the table 2 presents the results of a multivariate model (logistic regression) with all covariates in a same model. Therefore, we rightly referred to table 2 to support this sentence (“After adjustment, all causes of elevated B12 were more frequent in the High-B12 group, except for autoimmune or inflammatory diseases”).

Question 9: Page 6, table 3. I am not sure what this table adds. All relevant data are present to my opinion in figure 1. Maybe this table could be put in supplementals?

Response: We hope that we brought a clear answer to this comment in the question 4. The message delivered by the figure 1 and the table 3 were different. The figure 1 helped to validate the threshold-effect and to identify the best thresholds for non-metastatic and metastatic cancers as the table 3 showed the frequency of solid cancers increasing with the B12 level, notably above 1250 ng/L. In conclusion, we thought that figure 1 and table 3 were both important: the figure 1 certified a rigorous methodology and the table 3 brought a clear message for clinicians. However, if the reviewer or the editor still found that table 3 is unnecessary, we agree to place it in supplementary materials.

We did our best to modify our manuscript according to the comments of the reviewer and we hope that the answers will suit.

                Yours sincerely,

Geoffrey Urbanski, MD, MSc, PhD student

Department of Internal medicine, Angers University Hospital

4 rue Larrey, 49000 Angers, France

urbanskigeoffrey@gmail.com

Tel: +33 (0)2 41 35 40 03

Fax: +33 (0)2 41 35 49 69

Reviewer 2 Report

This work is an interesting population study.   

I think the Authors are enough experts in regard to the link between elevated plasma level of  vitamin B12 and solid cancers. Previously, few studies evaluated the strength of this association and, recently, it has already been documented by two retrospective cohort studies, based on health national registries (Danish and British). However, the design of these studies did not allow for the adjustment for other conditions associated with elevated B12 plasma levels.

The objectives of this study were to confirm this association after the adjustment for all causes of elevated B12 and to study the variations according to the increasing B12 level, the type of cancers and the presence of metastases. Then, this study, lasted 10 years, is an overcoming of the previous ones and certainly adds value to what has already been highlighted by analysis of the Danish and British registers.

In the current study, carried out on 10,573 patients who underwent B12 assay, elevated B12 plasma levels (≥1000 ng/L) were found in 785 patients (7.4%). In this regard I have a question for the Authors: how they consider this value (7.4% of enrolled patients)? Is it significant? Is it a poor result? Is it high? Please, answer these questions in the "Discussion" section, the common Reader could be curious. By continuing, these 785 patients were compared with 785 control subjects having normal level of B12<1000 ng/L > (≤1000 ng/L). Data analysis confirms the results of previous studies by showing an association between solid cancers and elevated B12. Moreover, this study highlights that this association persists after adjustment for other causes of elevated B12. And more, adjusted OR was 2.0 [95% CI: 1.2-3.2] for non-metastatic cancers and 4.2 [95% CI: 2.7-6.6] for metastatic cancers with a threshold of 1000 ng/L defining the elevated B12, and the strength of the association increases with the increasing B12 level, in particular in cases of metastases. The solid cancer and metastatic sites most associated with elevated B12 were pancreas, colon/rectum, lungs, prostate, urothelium, and bone and liver metastases. Contrary to results from previous registries’ studies, this study did not confirm the association of elevated B12 with liver cancers after adjustment for liver diseases.

From the methodological point of view the study is adequately appropriate and suitable to scientific rigour and ethical principles. The study design process is valid, both inclusion and exclusion criteria are compliant, as well as data collection; the statistical analysis seems valid; purposes and scientific contents are supported by valid References; editorial structure, figure and tables are good. The results are clear and the “Discussion” section is well structured and interesting.

In summary, the Authors conclude that unexplained elevated plasma levels of vitamin B12 should question the possibility of a solid cancer. If these data will be confirmed in other studies, they would improve pathway but, overall, prevention, diagnosis and follow-up (cancer biomarkers? note that "biomarkers" is a keyword of the manuscript) of some solid cancers. I would expect stronger focus on these aspects by the Authors, in the "Disussion" section. In this way, the effort of the Authors will be remarkable and I think this research will be a useful tool for other large scale studies and it will stimulate other Researchers.  

Moreover, I suggest the Authors to add a "Conclusion" section, at the end of the manuscript, briefly emphasizing the most important aspects of this study.

However, in my opinion, the manuscript deserves to be published; I only suggest the Authors to take into account my suggestions and observations.

Author Response

Response to reviewer 2: revised manuscript jcm-714905

February, the 4th, 2020

Dear Editor,

Dear Reviewer,

                We wish to thank the reviewer for the analysis of our work and for the helpful comments. We are thankful to let us submitting a revised manuscript for this study that highlights the association between solid cancers and elevated plasma vitamin B12, even after adjustment for other known causes of elevated B12.

                Please find below the answers to the comments in order to clarify and improve the quality of our manuscript.

Reviewer 2:

Comment 1: This work is an interesting population study. I think the Authors are enough experts in regard to the link between elevated plasma level of vitamin B12 and solid cancers. Previously, few studies evaluated the strength of this association and, recently, it has already been documented by two retrospective cohort studies, based on health national registries (Danish and British). However, the design of these studies did not allow for the adjustment for other conditions associated with elevated B12 plasma levels. The objectives of this study were to confirm this association after the adjustment for all causes of elevated B12 and to study the variations according to the increasing B12 level, the type of cancers and the presence of metastases. Then, this study, lasted 10 years, is an overcoming of the previous ones and certainly adds value to what has already been highlighted by analysis of the Danish and British registers.

Response: We thank the reviewer for summarizing our work as well.

Question 1: In the current study, carried out on 10,573 patients who underwent B12 assay, elevated B12 plasma levels (≥1000 ng/L) were found in 785 patients (7.4%). In this regard I have a question for the Authors: how they consider this value (7.4% of enrolled patients)? Is it significant? Is it a poor result? Is it high? Please, answer these questions in the "Discussion" section, the common Reader could be curious.

Response: We thank the reviewer for highlighting this aspect that we pointed out in the first version of the manuscript and that we then withdrew to limit the word count. In our study, plasma vitamin B12 ≥1000 ng/L was observed in 7.4% of patients. This result was slightly lower than that of previously published studies involving inpatients with similar inclusion criteria, ranging from 10 to 15% (Jammal et al. Rev Med Interne. 2013;34:337-41 in patients from emergency, internal medicine, acute geriatrics and medical intensive care departments, Chiche et al. Rev Med Interne. 2008;29:187-94 in patients from internal medicine department). Population-based studies found a slightly smaller prevalence of elevated B12 in general population: 3.5% of the population had a B12 >813 ng/L in the British registry (Arendt et al. Cancer Epidemiol Biomarkers Prev. 2019;28:814-821), and 6.6% of the population had a B12 >1084 ng/L in the Danish registry (Arendt et al. Cancer Epidemiol. 2016;40:158-65). The differences between these studies may be explained by variations in the recruitment (mean age, inpatients or outpatients, departments), and the threshold defining the status of elevated B12. The proportion of 7.4% in the manuscript was in accordance with another (prospective) study on vitamin B12 that we conducted with 38/528 (7.2%) patients with plasma vitamin B12 ≥1000 ng/L in our internal medicine department (unpublished data).

We think that the frequency of patients with plasma vitamin B12 ≥ 1000 ng/L in our study is significant when considering that this abnormality is an incidental finding (test performed to search a deficiency in most cases), and that the involved departments (Neurology, Geriatrics and Internal Medicine) had no specific recruitment of patients with liver or malignant diseases This highlights that studies on elevated vitamin B12 are of interest for clinicians because this abnormality is not uncommon. We feel necessary to assess the exact causes of elevated plasma vitamin B12 to help clinicians to adequately explore it. As proposed with relevance by the reviewer, we comment this result in view of previous studies in the revised version of the manuscript (page 8, line 232-240).

Comment 2: By continuing, these 785 patients were compared with 785 control subjects having normal level of B12 (≤1000 ng/L). Data analysis confirms the results of previous studies by showing an association between solid cancers and elevated B12. Moreover, this study highlights that this association persists after adjustment for other causes of elevated B12. And more, adjusted OR was 2.0 [95% CI: 1.2-3.2] for non-metastatic cancers and 4.2 [95% CI: 2.7-6.6] for metastatic cancers with a threshold of 1000 ng/L defining the elevated B12, and the strength of the association increases with the increasing B12 level, in particular in cases of metastases. The solid cancer and metastatic sites most associated with elevated B12 were pancreas, colon/rectum, lungs, prostate, urothelium, and bone and liver metastases. Contrary to results from previous registries’ studies, this study did not confirm the association of elevated B12 with liver cancers after adjustment for liver diseases.

From the methodological point of view the study is adequately appropriate and suitable to scientific rigour and ethical principles. The study design process is valid, both inclusion and exclusion criteria are compliant, as well as data collection; the statistical analysis seems valid; purposes and scientific contents are supported by valid References; editorial structure, figure and tables are good. The results are clear and the “Discussion” section is well structured and interesting.

Response: We greatly thank the reviewer for these comments about our work.

Question 2: In summary, the Authors conclude that unexplained elevated plasma levels of vitamin B12 should question the possibility of a solid cancer. If these data will be confirmed in other studies, they would improve pathway but, overall, prevention, diagnosis and follow-up (cancer biomarkers? note that "biomarkers" is a keyword of the manuscript) of some solid cancers. I would expect stronger focus on these aspects by the Authors, in the "Disussion" section. In this way, the effort of the Authors will be remarkable and I think this research will be a useful tool for other large scale studies and it will stimulate other Researchers.   

Response: Despite more available knowledge concerning causes of elevated B12 by means of recent studies, there is still no codified approach for exploring incidental elevated B12 (Arendt JFH. et al. Clin Chem Lab Med. 2013;51 / Ermens AA. et al. Clin Biochem. 2003;36:585-590 / Serraj K. et al. Pres Med. 2011;40:1120-1127). Furthermore, no prospective study has evaluated the interest of the vitamin B12 assay as a screening test for solid cancers. As a consequence, we remained cautious about the place of vitamin B12 screening for solid cancers: the design of the study did not allow recommending vitamin B12 assay as a screening tool for solid cancer but aimed at guiding clinicians in the determination of causes of incidental finding of elevated vitamin B12.

In case of incidental finding of an elevated plasma vitamin B12, we think that a clinical examination should be completed with simple additional tests (blood count, blood creatinine level, liver function assessment, liver echography) to eliminate the elevated B12-related causes other than solid cancers. If no cause of elevated B12 is found after these first explorations, the examination should be repeated focusing on specific organs that we found more frequently associated with solid cancer. The relevance of diagnostic imaging in screening remains unknown, and highly depends of the context (age, general condition, symptoms…).

Furthermore, our study observed an increasing association between vitamin B12 elevation and cancer extension. At this stage of knowledge concerning elevated vitamin B12, this finding raises many interesting questions about the place of the evolution of the vitamin B12 level during cancer treatment as a prognosis marker. So, raised questions would be: 1) what is the evolution of plasma vitamin B12 in patients with elevated levels at diagnosis during the 3 or 6 first months after starting solid cancer treatments? 2) is the evolution of the plasma B12 related to cancer evolution?

As rightly suggested by the reviewer, we added some comments in the “Discussion” section of the revised version of the manuscript (page 9 lines 310-318).

Comment 3: Moreover, I suggest the Authors to add a "Conclusion" section, at the end of the manuscript, briefly emphasizing the most important aspects of this study.

Response: As proposed by the reviewer, we added a “Conclusion” section at the end of the manuscript highlighting the key messages of our study.

Comment 4: However, in my opinion, the manuscript deserves to be published; I only suggest the Authors to take into account my suggestions and observations.

Response: We are very thankful to the reviewer for the encouraging comments. We hope that we responded to the questions with clarity and precision.

We did our best to modify our manuscript according to the comments of the reviewer and we hope that the answers will suit.

                Yours sincerely,

Geoffrey Urbanski, MD, MSc, PhD student

Department of Internal medicine, Angers University Hospital

4 rue Larrey, 49000 Angers, France

urbanskigeoffrey@gmail.com

Tel: +33 (0)2 41 35 40 03

Fax: +33 (0)2 41 35 49 69